# Insights from IgE Immune Surveillance in Allergy and Cancer for Anti-Tumour IgE Treatments

**DOI:** 10.3390/cancers13174460

**Published:** 2021-09-04

**Authors:** Alex J. McCraw, Jitesh Chauhan, Heather J. Bax, Chara Stavraka, Gabriel Osborn, Melanie Grandits, Jacobo López-Abente, Debra H. Josephs, James Spicer, Gerd K. Wagner, Sophia N. Karagiannis, Alicia Chenoweth, Silvia Crescioli

**Affiliations:** 1St. John’s Institute of Dermatology, School of Basic & Medical Biosciences, King’s College London, London SE1 9RT, UK; alexa.mccraw@kcl.ac.uk (A.J.M.); jitesh.chauhan@kcl.ac.uk (J.C.); heather.bax@kcl.ac.uk (H.J.B.); chara.stavraka@kcl.ac.uk (C.S.); gabriel.osborn@kcl.ac.uk (G.O.); melanie.grandits@kcl.ac.uk (M.G.); jacobo.lopez-abente@kcl.ac.uk (J.L.-A.); Debra.Josephs@gstt.nhs.uk (D.H.J.); G.Wagner@qub.ac.uk (G.K.W.); sophia.karagiannis@kcl.ac.uk (S.N.K.); 2School of Cancer & Pharmaceutical Sciences, King’s College London, Guy’s Hospital, London SE1 9RT, UK; james.spicer@kcl.ac.uk; 3Medical Biology Centre, School of Pharmacy, Queen’s University Belfast, 97 Lisburn Road, Belfast BT9 7BL, UK; 4Guy’s Cancer Centre, Breast Cancer Now Research Unit, School of Cancer & Pharmaceutical Sciences, King’s College London, London SE1 9RT, UK

**Keywords:** IgE, antibodies, cancer, immunotherapy, AllergoOncology

## Abstract

**Simple Summary:**

The growing field of AllergoOncology has illustrated potential for the use of IgE in cancer immunotherapy; however, there is still much to be explored within this field, particularly surrounding the links between IgE, allergy, and cancer. Exploring such links may provide useful insights to guide novel IgE-based strategies targeting cancer. Here, we summarise the existing data on both IgE in cancer epidemiology and tumour immunosurveillance, leading to the proposal of a new hypothesis, the combinatorial hypothesis, which attempts to encapsulate the complexity of the relationship between IgE-associated immune responses with cancer; and we discuss how these insights may shape the next generation of IgE-based therapeutics.

**Abstract:**

IgE, the predominant antibody class of the allergic response, is known for its roles in protecting against parasites; however, a growing body of evidence indicates a significant role for IgE and its associated effector cells in tumour immunosurveillance, highlighted by the field of AllergoOncology and the successes of the first-in-class IgE cancer therapeutic MOv18. Supporting this concept, substantial epidemiological data ascribe potential roles for IgE, allergy, and atopy in protecting against specific tumour types, with a corresponding increased cancer risk associated with IgE immunodeficiency. Here, we consider how epidemiological data in combination with functional data reveals a complex interplay of IgE and allergy with cancer, which cannot be explained solely by one of the existing conventional hypotheses. We furthermore discuss how, in turn, such data may be used to inform future therapeutic approaches, including the clinical management of different patient groups. With epidemiological findings highlighting several high-risk cancer types protected against by high IgE levels, it is possible that use of IgE-based therapeutics for a range of malignant indications may offer efficacy to complement that of established IgG-class antibodies.

## 1. Introduction

Since the advent of the first anti-cancer therapeutics, a major objective in the field of oncology has been the development of therapies capable of specifically targeting and killing cancer cells whilst preserving healthy cells. This ambition has in recent years driven a shift away from classical non-specific therapies such as radiotherapy and chemotherapy towards novel targeted therapies, amongst which cancer-specific monoclonal antibodies (mAbs) have risen to prominence [1,2]. Immunoglobulin G (IgG), the most prevalent human immunoglobulin in circulation, represents the majority of currently approved therapeutic mAbs, with the IgG1 isotype specifically forming up to 80% of current therapeutic IgGs [3,4]. A number of features—including favourable pharmacokinetics, long serum half-life, and well-described immune effector functions [5,6]—render IgG efficacious as an anti-cancer agent. However, limitations specifically affecting IgG-based antibodies such as low affinity for Fcγ receptors (FcγR); FcγR polymorphisms that influence IgG affinity and effector function potency; and engagement of the inhibitor receptor FcγRIIB, which dampens pro-inflammatory signaling, can affect their potency against specific cancers [7,8,9,10,11]. On this basis, there has been a push for novel means of overcoming these limitations, of which one approach has been the development of Ig-based cancer therapeutics of different antibody classes.

One such class is immunoglobulin E (IgE), which is commonly associated with the allergic response and immune responses against parasites and animal venoms [12,13]. IgE has been gaining traction as an alternative therapeutic, particularly with the growing field of AllergoOncology [14,15]; and work within this field has culminated most recently in the Phase 1 clinical trial of the anti-Folate Receptor α IgE MOv18 for advanced solid tumours (ClinicalTrials.gov Identifier: NCT02546921). A number of class-specific features of IgE, including its very high affinity for cognate Fcε receptors [16] and increased tissue residency through slow dissociations rates of receptor-bound IgE [12], are proposed to offer advantages in a cancer therapy setting. In addition, IgE lacks several of the limitations of IgG—no inhibitory receptors or polymorphisms influencing binding are described; and IgE can persist in tissues for extended periods of time without antigen engagement, which may be able to complement IgG surveillance in circulation [16,17]. Several studies of IgE class antibodies targeting different cancer-associated antigens support the notion that IgE immunotherapy may be able to complement IgG therapy treatments or even replace it in specific cancer settings [18,19].

Supporting a role for a powerful immune surveillance provided by IgE, links between IgE immune responses and protection against malignancies were first reported in the 1960s [20,21,22,23], and numerous studies point towards an involvement of IgE in natural anti-tumour immunosurveillance [24,25,26,27,28,29,30,31]. Despite this, the exact nature of the relationship between IgE, IgE-mediated conditions such as atopy and allergy, and cancer risk remain unclear. In this review, we therefore summarise the existing literature on putative roles of IgE class antibodies in natural immunosurveillance that may influence cancer risk; and we consider how this may translate to increased efficacy for IgE as a novel anti-cancer therapeutic modality.

## 2. Epidemiological Evidence Underlying IgE Immune Surveillance and Links with Cancer

### 2.1. The Four Hypotheses of IgE and Cancer

IgE represents a powerful immune activating antibody class, with known roles in allergy and features theoretically able to translate to heightened efficacy in a cancer therapy setting. Interestingly, links between IgE or IgE-mediated diseases and malignancy have been reported as far back as the 1960s, with early studies finding decreased cancer risk associated with the presence of allergy [20,21,22] and decreased prevalence of atopy in cancer patients [23]. Since then, numerous studies have been conducted, each examining epidemiological links with cancer—either looking at allergy and atopic disease more broadly, or specifically examining links between levels of IgE in circulation and cancer risk. Despite this, the exact nature of these relationships and their effects on cancer development remain debated, and contradicting evidence has rendered it difficult to draw conclusions.

A number of hypotheses, summarised in Table 1, have been proposed to explain the complex relationships between IgE, allergy, and cancer, the classical trinity of which is: (a) the chronic inflammation hypothesis, being that atopic inflammation drives oxidative damage and subsequent gene mutation, leading to heightened cancer risk; (b) the immune surveillance hypothesis, stating that atopy is a reflection of general enhanced immune responsiveness, which subsequently provides increased capability for removing dysregulated pre-cancerous cells; and, finally, (c) the prophylaxis hypothesis proposing that symptoms of the allergic response, such as coughing and sneezing, can decrease cancer risk by expelling potential carcinogens, promoting tissue repair, and driving behavioural avoidance of potential carcinogens [32,33]. A fourth hypothesis has subsequently been proposed—the Th2 skewing hypothesis, which claims atopy drives an inappropriate skewing towards T-helper 2 (Th2) responses over potentially tumour-eradicating T-helper 1 (Th1) immune responses, thereby enabling an immunosuppressive environment permissive of cancer development at sites of atopic inflammation [34].

However, despite a large body of evidence at hand to investigate relationships between IgE, allergy/atopy, and cancer risk, no conclusive stance has yet been reached on the exact nature of these relationships. One challenge, naturally, is comparison between studies—differing methodology, limited data on how atopy is defined and/or confirmed, and a lack of consideration of other factors potentially affecting cancer development, including smoking and family history. Collectively, these render it difficult to collate and compare results; and for a large number of cancers, results are too heterogenous with insufficient number of patients per study available to draw accurate conclusions. Another issue is the black-or-white stance often taken to explain the relationship of IgE and atopy with cancer, with the assumption often made that these relationships fall under clearly defined boundaries translating to increased or decreased risk overall. In reality, this relationship may be far more complex and likely reflects a combination of all four of the above proposed hypotheses (Figure 1). Inflammation driven by allergic responses, particularly in chronic conditions such as asthma and atopic dermatitis (AD), alongside Th2 skewing serves to increase cancer risk at sites of allergic response, but in turn, the increased immunosurveillance and prophylaxis afforded by allergy and its symptoms serve to protect against cancer at sites distant to that of the allergic disease’s primary manifestation. Such a combinatorial explanation may aid in understanding the complex interplay of IgE and its manifestations with cancer.

### 2.2. Chronic Inflammation and Cancer Risks Presented by Chronic Allergic Diseases

The chronic inflammation hypothesis presumes that symptoms of allergy drive inflammation at sites of the primary allergic manifestation and, therefore, cancer risk is subsequently increased at those sites, although not necessarily overall. Recent work by Hayes et al. [36] provided evidence in support of this hypothesis, with the demonstration that skin inflammation in mice enhanced polyclonal IgE levels, leading to basophil recruitment to these cutaneous sites, and increased epithelial cell growth and differentiation driven via FcεRI signalling in basophils. These led to the outgrowth of pre-cancerous epithelial cells harbouring oncogenic mutations and subsequent tumour growth [36]. In turn, mice that were IgE-deficient showed a substantially altered microenvironment at the skin, with decreased inflammation-driven hyperplasia; and mice deficient in FcεRI or FcεRI effector cells, such as basophils, were protected against tumour development, indicating putative roles for both IgE and FcεRI signalling in this process. Although not a direct model of atopic disease, this study points to a link between chronic local IgE-based inflammation and cancer risk, mandating the need for future studies to confirm this link in allergic diseases.

When looking specifically at epidemiological evidence, two strong relationships become clearly apparent within the data—a link between asthma and a substantially increased risk of lung cancer [37], and links between atopic dermatitis and increased risk of non-melanoma skin cancers (NMSC) [38], observations that appear to validate the chronic inflammation hypothesis as well as the Th2-skewing hypothesis. For lung cancer, a number of studies noted an increased risk associated with asthma history, even when accounting for factors such as smoking [37,39,40,41,42,43,44]. In contrast, when looking at other conditions—including hay fever [45], allergic rhinitis [46], eczema [38,47,48], or overall allergy history [37,39]—the inverse becomes true: lung cancer risk is decreased. Several meta-analyses conducted on the subject have ascertained this pattern, with some minor disagreement on the impact of smoking [37,49,50]. Of particular interest, Rosenberger et al. noted that when specifically stratifying results by age of asthma onset, lung cancer risk was significantly increased for those diagnosed with asthma at age 20 or later; potentially suggesting that unmanaged asthma symptoms greatly increase risk, likely through chronic inflammation [49]. However, observations that asthma is protective against several other cancer types separate from the lung [41,51,52,53,54] indicate a more complex relationship. Furthermore, the heterogeneous nature of asthma may contribute to the contradicting reports. For example, many studies do not distinguish between IgE-dependent and non-IgE-dependent asthma. Future research, which will consider the different asthma phenotypes and endotypes, might therefore be useful to unravel this complex relationship.

A similar pattern emerges when looking at skin cancer: The risk of basal cell carcinoma (BCC) and squamous cell carcinoma (SCC) is typically increased with a history of AD [38,55,56,57,58], but decreased when looking at other atopic conditions, including allergic rhinitis [46,51,59,60]. Interestingly, however, melanoma risk seems to be consistently decreased in conjunction with a history of allergy, in comparison to NMSC [33,37,43,56,58]. As with asthma, atopic dermatitis appears protective at sites distal to the site of inflammation, including lung cancer [38], colorectal cancer [51], and glioma [51,61,62], once more suggesting that risk only arises at sites directly exposed to inflammatory mediators.

Taken together, both the findings for lung and skin cancers suggest that chronic inflammation at sites of atopic disease may indeed play a role in driving site-specific cancer risk. However, in turn, the heightened immune responsiveness, as dictated in the immunosurveillance hypothesis, may act to prevent cancer occurrence at sites distant to the primary allergy, as indicated by the otherwise protective effects of asthma and AD observed on other cancer sites. As such, allergy and atopy may act as a double-edged sword, with both detrimental and protective effects in the context of cancer.

### 2.3. The Involvement of Prophylaxis and Th2 Skewing

A major component of the prophylaxis hypothesis is the presumption that the allergic symptoms themselves—such as the act of coughing and sneezing in response to allergen exposure—are themselves protective, by acting to expel potential carcinogens before they can exert damage [34]. In support of this, a decreased risk of upper gastrointestinal cancers—including oesophageal, stomach, and larynx cancers—was noted in conjunction with respiratory allergies. A number of studies have found a protective effect of any allergy or respiratory atopic diseases such as allergic rhinitis and hay fever against oesophageal cancer [45,51,63,64]; asthma against stomach cancer [41,47]; and inverse associations between asthma and larynx cancer, as well as allergic rhinitis and both hypopharyngeal and tongue cancers [41,51,63]. Under the prophylaxis hypothesis, it may be expected that asthma should therefore protect against lung cancers: The fact that the opposite is true lends further credence to a proposal for the intersection of multiple hypotheses.

Another underlying assumption of the prophylaxis hypothesis is that suppression of symptoms would mitigate this protection, and, indeed, several studies report evidence in support of this assumption. Such a relationship becomes apparent when looking at, for instance, glioma—a site-specific cancer that consistently shows decreased risk associated with allergy and atopy [53,65]. When antihistamines are used by those with a history of allergy/atopy, this protective effect is lost and, in some cases, risk may be increased [66,67]. Of interest, however, a number of studies have showed a contradictory decreased risk for glioma associated with use of antihistamines [68,69], which may potentially further validate the Th2-skewing hypothesis, as there has been some suggestion that Th1 responses may be protective against glioma [53]. However, a direct effect of antihistamines on the CNS cannot be excluded. Different types of antihistamines may differ in their penetration into the CNS, which could also explain the discordance reported by individual studies on the association of glioma with antihistamines [70]. The same study, however, also speculated that Th2 cytokines—such as interleukin (IL)-4 and IL-13—may hold protective roles against glioma via humoral immunity, including antibody production by stimulated B- cells. These speculations are supported by the understanding that peritumour inflammation is capable of altering the blood–brain barrier to permit immune cell infiltration [53].

There is indeed considerable evidence to support a role for Th2 cytokines in protecting against gliomas, including from animal model studies demonstrating roles for IL-4 and activated eosinophils in glioma suppression [71,72] and overexpression of IL-4 driving rejection, or regression, of glioma in rats [73,74]; as well as multiple associations made between glioma risk, progression, and polymorphisms in IL-4R and IL-13 genes [75,76,77]. It has been suggested that immune regulation in the central nervous system could potentially be mediated through IgE responses to allergens acting to minimise risk of central nervous system (CNS) tissue damage from inflammatory cell-mediated immune responses; and, in turn, the Th2 skewing in atopic individuals may in fact be more adept at protecting against brain tumours [78].

## 3. Roles of IgE in Tumour Immunosurveillance

### 3.1. IgE and Immunosurveillance: Illustrating the Potential for IgE-Based Therapeutics

Although IgE is arguably much maligned for its roles in allergy, roles in protecting against both parasites and animal venoms have been ascribed [12,13], but the question of whether this is enough to warrant its conservation across mammals is debated [30]. Whilst the exact role and function of endogenous tissue IgE in healthy individuals is unclear, it is thought to be involved in innate tissue defence, particularly against substances threatening tissue integrity [33,79], with suggestions that spontaneous IgE production may form an element of innate tissue defence [80,81]. IgE responses typically occur within epithelial tissues, barrier surfaces continuously exposed to environmental threats, some of which may be carcinogenic [79]; and IgE itself is able to engage and activate a unique repertoire of effector cells, corresponding to a powerful immune response (Figure 1).

Along this line of thought, another emerging role for IgE is proposed to be natural immune anti-tumour surveillance, supported by both epidemiological evidence as well as multiple functional and animal model studies. This feeds into the third major theory explaining relationships between IgE, atopy and cancer: immunosurveillance, providing a corresponding decreased malignancy risk.

Studies utilising animal models support the existence of tumour-protective functions for IgE [27,82,83,84]. Amongst these, several have demonstrated the validity of using monoclonal IgE antibodies engineered to recognise cancer antigens. Murine IgE antibodies recognising a mammary tumour virus antigen and a colorectal cancer antigen were each able to restrict the growth of subcutaneous tumours at lower doses than required for the equivalent IgG [24,25]; and, similarly, a human anti-HER2 IgE both restricted intraperitoneal tumour growth and prolonged the survival of mice transgenic for human FcεRIα. Introduction of this antibody was subsequently well-tolerated in cynomolgus monkeys [82]. IgE has also been shown to cross-react with non-human primate FcεRI albeit with faster dissociation from cynomolgus monkey versus from human FcεRI effector cells [83]. These differences were reflected in different effector mechanisms and immune activating profiles compared to IgG. Therefore, careful consideration of pre-clinical models may be required for toxicological evaluations of IgE immunotherapeutic candidates [85]. Using the same FcεRIα transgenic mouse model, a chimeric human/mouse IgE antibody specific for the prostate-specific antigen prolonged survival when these mice were challenged with prostate cancer cells, with investigation of the mechanisms of action revealing enhanced dendritic cell (DC) cross-presentation triggering T cell responses [29]. Further studies demonstrated additional involvement of T-cells, with depletion of CD8+ T-cells sufficient to impair IgE-dependent tumour protection, highlighting the involvement of these cells downstream of IgE-FcεRI signalling [30].

Separately, IgE has been shown to be capable of re-educating alternatively activated pro-tumour M2 macrophages towards a pro-inflammatory state and of priming all macrophage subsets towards an anti-tumour function, suggesting potential roles for the use of IgE in targeting tumour-associated macrophages (TAMs) [31]. TAMs are generally associated with poor prognosis in solid tumours, with characteristics demonstrated to enable tumour growth, and are an emerging target for cancer immunotherapy [31]. Research has suggested that IgE may be capable of inducing more mature pro-inflammatory phenotypes in macrophages, resembling the M1 phenotype, capable of driving anti-tumour functions [19,31,86]. Similar observations have been made in human monocytes, with IgE capable of activating monocytes from patients with ovarian cancer and from healthy subjects into pro-inflammatory phenotypes capable of mediating cytotoxicity of tumour cells [86].

Several lines of evidence indicate not only roles for IgE but also for its Fc receptors in tumour immunosurveillance. CD23, the low-affinity IgE Fc receptor, has been implicated in cancer patient outcomes. Olteanu et al. [87] and Linderoth et al. [88] noted a significantly higher survival rate in patients with lymphoma whose cancer cells expressed CD23, compared to those that were CD23-negative: not only suggesting that IgE binding to tumour cells may drive tumour inhibition or killing, but also further suggesting a role for IgE in circulatory immunosurveillance. Karagiannis et al. [28] indicated a putative role for CD23 expressed on IL-4 stimulated monocytes in driving anti-tumour IgE antibody-dependent cell phagocytosis of ovarian tumour cells; whilst Ye et al. [89] demonstrated through cell culture models that IgE interactions with CD23 expressed on colon cancer cells could trigger cancer cell apoptosis. Taken together, these data suggest that IgE antibodies may provide local and long-lasting anti-tumour defence, with putative roles for CD23 in some of these processes; as well as putting forward the possibility that even the low levels of IgE in circulation in healthy individuals may aid in this immunosurveillance function.

Such observations all feed into the growing field of AllergoOncology [14], which focuses on studying the links between IgE and Th2 immunity with allergy and cancer. Research in this field looks towards exploiting natural anti-tumour functions of IgE for therapeutic benefit, acting to supplement the principally IgG-dominated area of cancer therapeutics. Indeed, there are suggestions that IgE may not only be able to complement IgG therapy, but in some cases, may offer superior benefit—for example, in some model systems, IgE antibodies were capable of mediating antibody-dependent cell-mediated cytotoxicity (ADCC) against corresponding cancer cells at greater superiority compared to a corresponding IgG1 (70% vs. 30% in ADCC/ADCP assays) [18,90]. Furthermore, an IgE-trastuzumab was able to directly affect tumour cell viability in the absence of effector cells, putting it on par with IgG-trastuzumab, and lending support to the notion that anti-tumour IgE antibodies are as capable as, if not more than, IgG-based therapeutics in delivering direct blockade of cancer growth and survival signals [18]. Finally, the chimeric mouse/human monoclonal IgE, MOv18 is presently undergoing a Phase 1 trial for advanced solid tumours (ClinicalTrials.gov Identifier: NCT02546921). MOv18-IgE is directed against the tumour-associated antigen folate receptor alpha (FRα), which is expressed in upwards of 70% of ovarian carcinomas, as well as other tumour types [91]. MOv18-IgE has been confirmed to have both ADCC and ADCP capabilities mediated by FcεRI and CD23, respectively [26,28], and has been shown in vitro to drive ovarian cancer cell death by activating immune effector cells such as monocytes, macrophages, and mast cells [92]. MOv18-IgE’s primary mechanisms have been established as tumour cell killing via ADCC and ADCP through activation of IgE effector cells, and, across three models of cancer, MOv18-IgE proved more effective than its IgG counterpart [16]. Importantly, anti-tumour IgE antibodies directed against cancer-associated antigens have been shown to re-activate tumour-associated and alternatively activated macrophages towards pro-inflammatory states [19,31,86]. This harbours the potential to reverse immune suppressive forces and stimulate the tumour microenvironment against cancer cells [93,94]. Combined, these mechanisms may distinguish this class from the commonly used IgG isotypes currently used in the clinic, at least in the context of certain solid tumours where IgG treatments have historically been ineffective.

Although the MOv18-IgE clinical trial is currently ongoing, preliminary results suggest treatment is well tolerated, with high-risk side effects such as anaphylaxis predictable via ex vivo assays such as the basophil activation test (BAT), and a maximum tolerable dose has not yet been reached. Anti-tumour activity has been observed at doses of 0.7 mg, with some evidence of reduction of peritoneal metastases and with a decrease in CA125 serum marker levels meeting the Gynecologic Cancer InterGroup response criteria [95]. Combined with increasing evidence of anti-tumour functions for IgEs in the pre-clinical models, these results so far support both the safety and the efficacy of IgE-based cancer therapeutics, even with antibody administration at a fraction of doses required for conventional antibody therapeutics to achieve anti-tumour effects [95].

### 3.2. Impact of IgE Levels on Malignancy Risk

Whilst the majority of epidemiological studies have looked more broadly at allergy and atopic diseases as a whole in the context of cancer, several have more specifically investigated the impact of IgE levels (either serum total or allergen-specific) themselves on cancer risk and mortality. As with allergy and atopy, a general observation consistent across the majority of studies is that higher total serum IgE levels correlate to a decreased cancer risk, either overall [32,33,46] or by site-specific cancer [78,96,97,98]. In cancer patients themselves, there is evidence to suggest that higher IgE levels may correlate to a prolonged survival compared to those with low or even average levels [99,100], and functional data lends further credence. In separate studies, Nigro et al. [30] and Singer et al. [101] each employed mouse models to represent the high-IgE levels typically observed in atopic patients, as well as low- or IgE-deficient mouse models; with both observing an enhanced survival in the IgE-high mice compared to wild-type (WT). Complete lack of IgE response was sufficient to enhance tumour growth and decrease survival compared to WT mice, even with prior immunisation against tumour cells [30]. Furthermore, Singer et al. noted that low-IgE mice could benefit significantly from provision of immunotherapy, allowing them to reach survival rates on par with the high-IgE mice, whilst immunotherapy did not appear to further benefit the high-IgE group [101]. Taken together, this suggests an innate protective effect of high IgE levels from which atopic patients may benefit.

Although epidemiological studies focusing specifically on IgE levels in humans do suggest an overall protective effect of high serum IgE against general cancer [32,33,46], as well as towards specific cancers [33,78,98,102], there remains ongoing debate over the exact level of total IgE concentrations beyond which there is protection against cancer. Interestingly, and in line with increased cancer growth reported by Singer et al. in low-IgE mouse models [101], it has been shown that IgE immunodeficiency—typically IgE serum levels of <2.5 kU/L—appears to consistently present an increased risk of malignancy in adults and children. Across a series of studies, Ferastraoaru et al. reported that both risk and rate of prior cancer diagnosis was higher in IgE-deficient patients compared to their non-IgE-deficient counterparts [103,104,105]. A recent study demonstrated that, similar to IgE-deficient adults, IgE-deficient children were at increased risk for malignancy compared to those without IgE-deficiency, and that very low or absent IgE titres are a potential risk factor for the development of malignancy [105]. Similarly, their studies conducted in 2017 and 2018 showed that adults with IgE levels below 2.5 kU/L had higher rates of a prior diagnosis of any type of malignancy compared to those who had levels above 2.5 kU/L [103,104]. There was, however, some controversy over which tumour types were more prevalent in the IgE-deficient group, with Ferastraoaru and colleagues (2017) noting an increased prevalence of lymphoma and multiple myeloma in the IgE-deficient cohort, supporting previous findings from other groups [106,107,108]. In a subsequent study (2018), however, it was suggested that solid tumours were the most frequent in IgE-deficient patient groups [103]. As it stands, it may simply be that IgE deficiency is associated with increased risk of both solid and blood cancers arising from a loss of tumour immunosurveillance function. When looking at data patterns surrounding which cancer types are afforded protection by allergy or atopy, a significant proportion appear to be solid tumours—in line with the understanding that IgE may likely aid in tissue immunosurveillance. When it comes to haematological and lymphatic cancers in relation to allergy and atopy, findings are variable but there appears to be a consistent observation that IgE deficiency correlates to an increased risk of blood cancers, particularly lymphomas [96,104,105,106,107,109]. Although IgE is generally assumed to be predominantly sequestered in tissues, these findings may potentially also highlight a role for IgE in circulatory immunosurveillance.

Parallel to both these observations and other reports of the impact of allergy medication, such as antihistamines, on cancer risk, there has been investigation into whether use of the anti-IgE therapeutic omalizumab may present increased cancer risk, with accompanying FDA drug information indicating such concerns [110]. Omalizumab is a recombinant humanised mAb specifically targeting circulating IgE, which has been shown to rapidly reduce both serum-free IgE and expression of high-affinity IgE Fc receptors [111]. As such, it has been proposed that omalizumab may increase malignancy risk by impairing natural anti-tumour IgE-mediated surveillance. Recent meta-analyses were ultimately unable to conclude any definitive link between omalizumab treatment and cancer [112,113]; however, nor were they able to rule it out. A number of limitations in studies conducted so far include length of follow-up, selection bias, and lack of pre-enrolment data. As with many aspects of the complex relationship of IgE with cancer, there remains a pressing need for long-term comparative studies specifically focused on malignancy risk, recurrence, and disease progression.

## 4. The Combinatorial Hypothesis: Addressing the Gaps

When looking at the data surrounding the relationship between IgE, allergic diseases and cancer risk, it can often be difficult to see the bigger picture. Existing evidence highlights that these relationships are far more complex than a simple increased or decreased cancer risk and that current hypotheses may fall short of truly capturing the complexity of these relationships. In turn, improving our understanding of this relationship may be key to in turn connect epidemiological data to functional activities of IgE. So, it may be proposed instead that a fifth alternate hypothesis exists, the combinatorial hypothesis (Figure 2), combining the four existing hypotheses of immunosurveillance, prophylaxis, chronic inflammation, and Th2 skewing into one, reflecting the complex nature of these relationships and the involvement of IgE. When examining both epidemiological and functional findings, it becomes apparent that data are not biased towards any particular hypothesis over another, and instead, suggests that all three of the classical hypotheses at least may be simultaneously true. The overwhelming evidence in support of anti-tumour and immunosurveillance functions of IgE suggests these functions represent a key role in addition to its anti-parasitic roles that may explain the evolutionary conservation of IgE across mammalian species. Such roles may also explain the prevalence of allergy and atopic disease in developed nations. With Th2 responses being considered to have originally evolved to combat extracellular parasites, it has been hypothesised that, due to lower rates of parasitic infections particularly in westernised society, these previously specialised Th2 responses now instead react to otherwise harmless environmental allergens [114]. Another approach may therefore be that these specialised IgE responses have gradually adapted to combat potential carcinogens, or that this function has existed concurrently with its anti-parasitic roles. As such, prevalence of allergy and atopic disease may not simply be activation of an otherwise defunct immune response, but rather, misdirection of a potent immune response intended to protect against potential carcinogens and other harmful environmental substances.

Separately to insights on how epidemiological and functional data may reflect novel IgE functions, evaluating IgE and the risk of malignancy may also offer guidance towards characterising therapies beneficial for specific patient groups. If one looks at the cancers protected against by the presence of allergy/atopy or high serum IgE levels, a number of site-specific cancers stand out, these including lung cancer (in the absence of asthma) [37,38,39,45,46]; colorectal cancer, particularly rectal cancer [37,46,51,64,115]; and pancreatic cancer [37,38,50,52,64,116,117], all of which seemed to be protected against by either the presence of allergy/atopic disease or high serum IgE levels. Another cancer overwhelmingly benefiting from the presence of allergy or high serum IgE is glioma [51,53,65,78,97,100,118,119,120], with additional evidence to suggest that high serum IgE may even prolong survival in glioma patients [78,100]. Therefore, it is possible that IgE could represent a potential novel therapeutic agent for the treatment of such cancers. Furthermore, such observations could help direct treatment guidelines when looking at those cancers that seem to be protected against by allergy/high IgE titres and those negatively influenced by low IgE titres, such as by recommending the avoidance of treatments affecting allergy symptoms or IgE levels, including omalizumab and antihistamines. In cancers such as glioma, however, it remains unclear how exactly allergy and IgE may be protective and, as such, further research is urgently needed to understand the specific glioma microenvironment and its interactions with immune responses [53], before specific IgE immunotherapies could be developed. Other tumour types protected against by IgE, however, may stand to greatly benefit from novel therapies: pancreatic cancer, for example, is characterised by 5-year survival rates as low as 10%, with little progress made in advancing patient survival [121]. These immune-privileged tumours, whilst often devoid of T cell infiltration, are characterised by the presence of IgE effector cells such as macrophages. Pancreatic cancer may thus present an opportunity for IgE immunotherapy, taking advantage of the long tissue retention and potential for prolonged immune surveillance of this class, and the ability of IgE to direct macrophage-mediated tumour cell killing, and potentially antigen presentation and repolarisation of tumour supportive M2-like macrophage subsets.

However, there remains a pressing need for long-term studies specifically designed to investigate the links of IgE and different allergic conditions with specific cancers in order to properly define these relationships. Despite promising evidence, several major limitations across a large majority of epidemiological studies render comparisons difficult and often result in great heterogeneity between conclusions. A large factor often lies in the classification and confirmation of allergy itself, with some studies relying on self-reporting of allergic diseases, whilst others only include patients whose allergic disease has been confirmed via physician. Even with the latter, differences in criteria are evident, including the use of IgE measurements and skin-prick test positivity for diagnosis. Furthermore, long-term studies in large populations lack longitudinal IgE measurements prior to cancer diagnosis. For those patients with a cancer diagnosis, concerns may be that either malignancy or different treatments for malignancy may affect IgE serum levels [32,33,122]. Additional considerations include known confounding variables, such as lifestyle, smoking, genetic and environmental factors, and family history, all of which would further affect current data. Additionally, differences in health care between countries can make accurate assessment difficult—for example, access to antihistamines without prescription in the UK, where their use may be missed both in studies relying on self-reporting and medical files. The provision of long-term studies taking such factors into consideration would therefore be of great benefit to the field and further aid in our understanding.

## 5. Conclusions

Evidence from collective epidemiological data will be key to gain real insights into health risks and to inform the clinical management of different patient groups, such as those with allergies, asthma, and several immunodeficiencies, as well as of different cancer patient cohorts. Not only may this knowledge help stratify patients—for example, it has been suggested that ultra-low IgE levels may act as a biomarker for cancer risk [114], which may mean that individuals with low IgE titres may benefit from increased cancer screenings—but in turn, it may also help shape the future of both IgE therapeutics and cancer therapeutics as a whole. Analysis of epidemiological data may stand to highlight tumour types such as pancreatic cancer, which potentially stand to benefit from IgE-based therapy.

As our understanding of both cancer and the roles of IgE within anti-tumour immunosurveillance grows, a growing area for novel, previously underappreciated approaches to treating certain malignancies is emerging. Informed and supported by mechanistic and epidemiological studies in large but also in specific patient cohorts, therapeutic targeted IgE, with its long tissue residency, direct tumour cell killing, and ability to repolarise tumour-supportive M2-like macrophages, likely represent a promising new means of tackling cancer and may improve survival in cancers with poor outcomes.

## Figures and Tables

**Figure 1 cancers-13-04460-f001:**
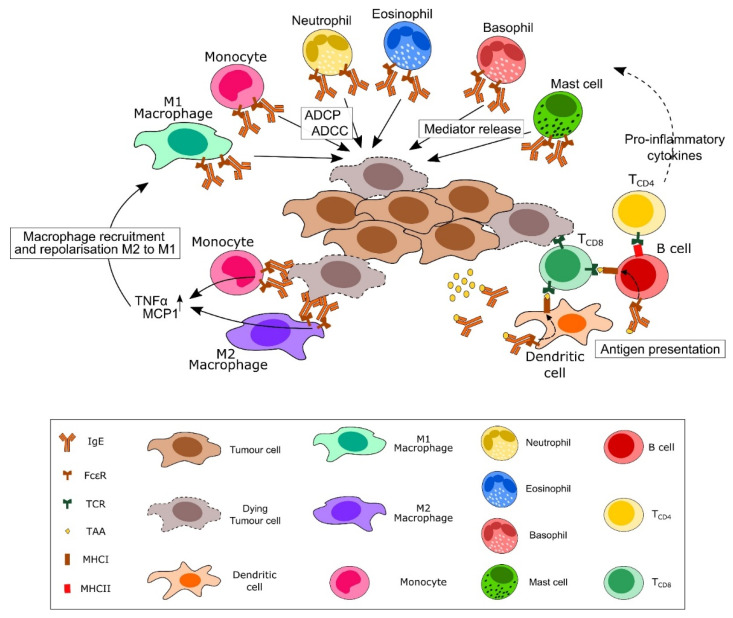
IgE and IgE therapy-mediated cellular immune surveillance against cancer. Direct and cell-mediated effects of IgE against tumour cells (shown in brown). IgE is able to engage with a unique repertoire of effector cells via the IgE receptors Fcε receptor I (FcεRI) and CD23/FcεRII. These interactions can drive potentially anti-tumour functions, including antibody-dependent cell-mediated cytotoxicity (ADCC) or phagocytosis (ADCP) and release of cytotoxic mediators. IgE engagement with monocytes and alternatively activated M2 macrophages can drive repolarisation to a M1-like classically activated phenotype, as well as increased macrophage recruitment to tumour sites. IgE can also contribute to antigen presentation via dendritic cells and B cells and activate both CD4 and CD8 T cells, further promoting a pro-inflammatory environment and tumour cytotoxicity.

**Figure 2 cancers-13-04460-f002:**
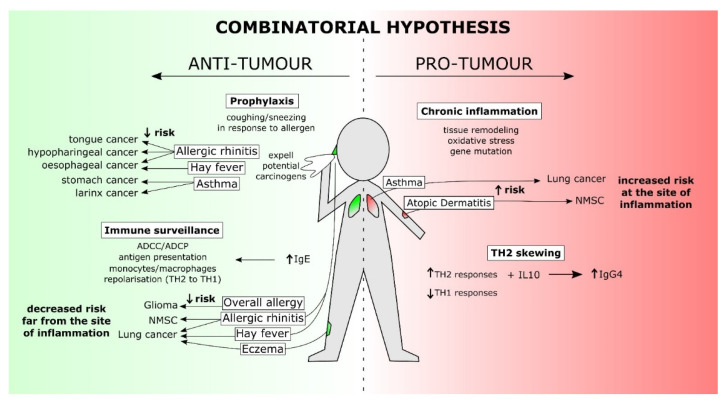
The combinatorial hypothesis accounts for the complex relationship between allergic diseases and an individual’s risk of developing a specific cancer. The combinatorial hypothesis takes into account different immunological conditions created within the context of specific local and systemic allergic diseases, which influence the risk for the development of different cancer types. This risk may be derived from four hypotheses thus far proposed to explain the nature of the relationship between IgE, allergy/atopy, and cancer. These hypotheses can in turn be divided into those that propose an anti-tumour outcome and those proposing a pro-tumour outcome. For anti-tumour outcomes, the prophylaxis hypothesis assumes that symptoms of allergy, such as coughing and sneezing, act to expel potential carcinogens before they can trigger tissue damage and inflammation. Reported decreased risk of cancers of the upper gastrointestinal tract associated with respiratory allergy is an example in support of this hypothesis. The second anti-tumour hypothesis is the immunosurveillance hypothesis, which states that allergy is reflective of a hyperactive immune system, which in turn is better equipped to defend against tumours. Supporting this, several cancers distant to the site of allergic inflammation are protected against by presence of allergy or high serum IgE. The pro-tumour hypotheses are the chronic inflammation hypothesis and the Th2 skewing hypothesis. The chronic inflammation hypothesis presumes that inflammation, driven by allergy, drives mutation of regulatory growth and tumour-suppressor genes, can increase cancer risk at sites of allergic challenge. This hypothesis is supported by increased lung cancer risk associated with asthma and increased non-melanoma skin cancer risk associated with atopic dermatitis. The Th2 skewing hypothesis, presumes atopy and chronic exposure to allergen challenge may drive inappropriate skewing towards alternatively activated Th2-based immune response over potentially anti-tumour Th1 or away from the typical Th2 (IgE-driven) responses, leading to an increased cancer risk at sites affected by atopy.

**Table 1 cancers-13-04460-t001:** Proposed hypotheses explaining relationships between allergy, IgE, and cancer.

Hypothesis	Description	Cancer Risk
Chronic Inflammation [34]	Allergy-induced events including inflammatory cell infiltration, tissue remodelling, and enzyme activation drive mutation of tumour suppressor genes, apoptotic proteins and other factors involved in regulation of cell growth, promoting growth of cancerous cells.Cancer risk will be increased at sites of chronic inflammation.	Increased
Immunosurveillance [34]	Allergy reflects general immune hyperresponsiveness; natural immunosurveillance is enhanced. Enhanced immunosurveillance is reflected by high serum IgE, potentially triggered against undiagnosed tumours instead of allergens. IgE/atopy drives activation of powerful effector cells capable of detecting and mounting responses against tumour cells	Decreased
Prophylaxis [35]	Physical effects of allergy such as coughing or sneezing act to expel potentially mutagenic or carcinogenic toxins before they can trigger malignancy. Allergy symptoms themselves are beneficial and not necessarily IgE.	Decreased
Th2 Immune Skewing [34]	Atopy drives an inappropriate skewing towards T-helper 2 (Th2)-based immune responses, diverting away from potentially tumour-eradicating inflammatory T-helper 1 (Th1) responses. High serum IgE concentrations as a result of Th2 skewing occupy receptors and prevent binding of anti-tumour IgE to Fcε receptor I (FcεRI)-expressing effector cells. Tissues affected by atopy are more sensitive to cancer development due to Th2 skewing.	Increased

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
