# Peer review of "Insights from IgE Immune Surveillance in Allergy and Cancer for Anti-Tumour IgE Treatments"

_cancers, 2021, doi:10.3390/cancers13174460_

Round 1
Reviewer 1 Report
This is a very well written review on IgE and cancer immunotherapy. They integrated the current hypotheses of IgE and Cancer to a new combinatorial hypothesis. They discussed the links between IgE, allergy, and cancer. Also they provided the insights to guide novel IgE-based strategies targeting cancer. Very comprehensive review on this fairly novel research field.
Author Response
Point by point response to reviewers
Reviewer 1
This is a very well written review on IgE and cancer immunotherapy. They integrated the current hypotheses of IgE and Cancer to a new combinatorial hypothesis. They discussed the links between IgE, allergy, and cancer. Also they provided the insights to guide novel IgE-based strategies targeting cancer. Very comprehensive review on this fairly novel research field.
Authors’ response:
We thank Reviewer 1 for the review of this manuscript and their kind comments.
Reviewer 2 Report
The manuscripts discusses a hypothesis concerning the role of IgE in tumor surveillance and/or its potential application as an anti-tumor therapy.
The concept is new and original. It is of utmost scientific and clinical significance.
The article is well-written and clearly shows the ideas presented by the Authors.
There are some minor flaws in the “introduction”.
- It should be borne in mind that asthma is not a homogenous disease. Therefore, differences between in incidence of lung tumor between asthma and other allergic diseases may be due to non-Th-2 type- and/or non-IgE-dependent asthma. In fact, the reference 37 addresses neither the issue of IgE nor that of Th-2 type inflammation. As an example the 36th citation in the meta-analysis (Ref 37) included “Asthma cohort: Those patients from the source population who had a recorded code for asthma at any time before or during the study period (1994–2001)”. Similarly, most of the other papers included in that meta-analysis did not address the asthma endotype/phenotype issues.
- Citation No 36 does not represent a model reflecting natural (atopic) allergy. It is therefore difficult to draw a conclusion from that study which concerns atopic diseases.
- The effects of anti-histamines on the development of glioma may depend on their direct action of on CNS which is different for different anti-histamines. In fact, the discordance reported by individual studies on the association of glioma with anti-histamines used for allergic diseases may be related to differences in penetration to CNS by individual anti-histamines.
Author Response
Point by point response to Reviewer 2
Reviewer 2
The manuscripts discusses a hypothesis concerning the role of IgE in tumor surveillance and/or its potential application as an anti-tumor therapy.
The concept is new and original. It is of utmost scientific and clinical significance.
The article is well-written and clearly shows the ideas presented by the Authors.
Authors’ response:
We thank Reviewer 2 for their positive comments and constructive feedback. In the following responses, we address your specific comments raised.
There are some minor flaws in the “introduction”.
- It should be borne in mind that asthma is not a homogenous disease. Therefore, differences between in incidence of lung tumor between asthma and other allergic diseases may be due to non-Th-2 type- and/or non-IgE-dependent asthma. In fact, the reference 37 addresses neither the issue of IgE nor that of Th-2 type inflammation. As an example the 36th citation in the meta-analysis (Ref 37) included “Asthma cohort: Those patients from the source population who had a recorded code for asthma at any time before or during the study period (1994–2001)”. Similarly, most of the other papers included in that meta-analysis did not address the asthma endotype/phenotype issues.
Authors’ response:
We thank the Reviewer for this observation. We agree about the importance of proper patient stratification when analysing comorbidities, which is indeed one of the limits of most of the past literature. We have now added to the text “Furthermore, the heterogeneous nature of asthma may contribute to the contradicting reports. For example, many studies do not distinguish between IgE-dependent and non-IgE-dependent asthma. Future research, which will consider the different asthma phenotypes and endotypes, might therefore be useful to unravel this complex relationship.” Lines 166-170.
- Citation No 36 does not represent a model reflecting natural (atopic) allergy. It is therefore difficult to draw a conclusion from that study which concerns atopic diseases.
Authors’ response:
We thank the Reviewer for this comment. We have now added to the text “Although not a direct model of atopic disease, this study points to a link between chronic local IgE-based inflammation and cancer risk, mandating the need for future studies to confirm this link in allergic diseases.”. Lines 147-150.
- The effects of anti-histamines on the development of glioma may depend on their direct action of on CNS which is different for different anti-histamines. In fact, the discordance reported by individual studies on the association of glioma with anti-histamines used for allergic diseases may be related to differences in penetration to CNS by individual anti-histamines.
Authors’ response:
We thank the Reviewer for this observation. We agree and we have now amended the text clarifying that “However, a direct effect of antihistamines on the CNS cannot be excluded. Different types of antihistamines may differ in their penetration into the CNS, which could also explain the discordance reported by individual studies on the association of glioma with anti-histamines." And we added the Church DS et al (2011) reference to support our statement. Lines 210-213